# Enhancement of an In Vivo Anti-Inflammatory Activity of Oleanolic Acid through Glycosylation Occurring Naturally in *Stauntonia hexaphylla*

**DOI:** 10.3390/molecules25163699

**Published:** 2020-08-13

**Authors:** Le Ba Vinh, Nguyen Thi Minh Nguyet, Liu Ye, Gao Dan, Nguyen Viet Phong, Hoang Le Tuan Anh, Young Ho Kim, Jong Seong Kang, Seo Young Yang, Inkyu Hwang

**Affiliations:** 1College of Pharmacy, Chungnam National University, Daejeon 34134, Korea; vinhrooney@gmail.com (L.B.V.); minhnguyet.usth@gmail.com (N.T.M.N.); liuye1995@naver.com (L.Y.); gaodan521361@hotmail.com (G.D.); yhk@cnu.ac.kr (Y.H.K.); kangjss@cnu.ac.kr (J.S.K.); 2Institute of Marine Biochemistry (IMBC), Vietnam Academy of Science and Technology (VAST), Hanoi 100000, Vietnam; ngvietphong@gmail.com; 3Mientrung Institute for Scientific Research, Vietnam Academy of Science and Technology (VAST), Thua Thien Hue 531600, Vietnam; hoangletuananh@hotmail.com

**Keywords:** anti-inflammatory effect, Lardizabalaceae, oleanane triterpene saponin, *Stauntonia hexaphylla*

## Abstract

*Stauntonia hexaphylla* (Lardizabalaceae) has been used as a traditional herbal medicine in Korea and China for its anti-inflammatory and analgesic properties. As part of a bioprospecting program aimed at the discovery of new bioactive compounds from Korean medicinal plants, a phytochemical study of *S. hexaphylla* leaves was carried out leading to isolation of two oleanane-type triterpene saponins, 3-*O*-[*β*-d-glucopyranosyl (1→2)-*α*-l-arabinopyranosyl] oleanolic acid-28-*O*-[*β*-d-glucopyranosyl (1→6)-*β*-d-glucopyranosyl] ester (**1**) and 3-*O*-*α*-l-arabinopyranosyl oleanolic acid-28-*O*-[*β*-d-glucopyranosyl (1→6)-*β*-d-glucopyranosyl] ester (**2**). Their structures were established unambiguously by spectroscopic methods such as one- and two-dimensional nuclear magnetic resonance and infrared spectroscopies, high-resolution electrospray ionization mass spectrometry and chemical reactions. Their anti-inflammatory activities were examined for the first time with an animal model for the macrophage-mediated inflammatory response as well as a cell-based assay using an established macrophage cell line (RAW 264.7) in vitro. Together, it was concluded that the saponin constituents, when they were orally administered, exerted much more potent activities in vivo than their sapogenin core even though both the saponins and the sapogenin molecule inhibited the RAW 264.7 cell activation comparably well in vitro. These results imply that saponins from *S. hexaphylla* leaves have a definite advantage in the development of oral medications for the control of inflammatory responses.

## 1. Introduction

Immunity is the body’s defense system against infectious agents or tumor formation. Proper magnitudes and durations of the responses are, however, critical for the health of the body as unnecessarily overactive or long-lasting immune responses can cause various immunological disorders compromising the quality of life. For example, while macrophages play a critical role in defense against bacterial infections, they can also cause inflammatory disorders when their responses persist for too long [1]. Thus, it is often necessary to suppress macrophage activation to prevent chronic inflammatory disorders. Medicinal herbs have been identified as rich sources of natural products with anti-inflammatory activities [2].

Diverse proinflammatory cytokines and mediators such as nitric oxide (NO), and tumor necrosis factor α (TNF-α) are mainly released by activated macrophages, which emerge in response to exogenous stress or endogenous signals such as external pathogen and damage-associated molecular patterns (DAMPs) [3]. These are key mediators to induce inflammatory disorders and may have potential as the therapeutic targets [4].

*Stauntonia hexaphylla* (Lardizabalaceae) has been used as a traditional herbal medicine in Korea, China and Japan for its analgesic and diuretic properties [5]. The genus *Stauntonia* produces various phytochemicals, such as oleanane triterpenoids [6], phenolic glycosides [7] and triterpene saponins [8], which exhibit diverse pharmacological effects including anti-HIV-1 protease [9], anti-gout [10,11], anti-diabetic [12] and anti-osteoporotic [13] effects. In the previous study, we had identified several triterpene glycosides that inhibited the activation of an established macrophage cell line called RAW 264.7 in vitro [14,15]. Previously, oleanane triterpene saponins have shown anti-inflammatory inhibitory effects, such as aster saponin C2 [16], and kalopanaxsaponin B [17]. In this study, we identified two major derivatives of oleanolic acid saponins and examined their pharmacological activities for inhibition of the activation of the RAW 264.7 cell line in vitro, relative to those of oleanolic acid aglycon (Figure 1). In this study, the preliminary comparison of anti-inflammatory activity of these saponins with their aglycon was also discussed. Additionally, to investigate the in vivo efficacies of the saponins and the aglycon, we performed an animal experiment widely used for determining macrophage-dependent immune response, i.e., the lipopolysaccharide (LPS)-induced mouse sepsis model, administered orally with either the saponin or the aglycon beforehand [14,15,18].

## 2. Results

### 2.1. Structure Identification of Isolated Compounds

Compound **1** was obtained as a white amorphous powder, with [α]d^20^ + 7.05 (*c* = 1.5, methanol (MeOH)). The infrared (IR) spectrum showed the presence of ester (1728 cm^−1^), olefin (1066 cm^−1^) and hydroxyl (3398 cm^−1^) groups. The molecular formula of compound **1** was deduced as C_53_H_86_O_22_ based on the electrospray ionization mass spectrometry (ESI MS) ion peak at *m*/*z* 1092.5 [M + NH_4_]^+^ (calcd. for C_53_H_90_NO_22_^+^, *m*/*z* 1092.5) and the ion peak at *m*/*z* 1119.5 [M + HCOO^–^]^–^ (calcd. for C_54_H_87_O_24_^−^, *m*/*z* 1119.5). The proton nuclear magnetic resonance (^1^H-NMR) data of **1** exhibited the typical signals of seven tertiary methyl groups at (*δ*_H_ 0.90, 0.91, 0.92, 1.04, 1.10, 1.21 and 1.25, each 3H, s) and four anomeric protons at *δ*_H_ 6.22 (d, *J* = 8.0 Hz, H-1′′′), 4.94 (d, *J* = 5.0 Hz, H-1′), 4.99 (d, *J* = 7.5 Hz, H-1′′) and 5.15 (d, *J* = 7.5 Hz, H-1′′′′) showing correlations in the heteronuclear single-quantum coherence (HSQC) spectrum with their four anomeric carbon signals at *δ*_C_ 96.2, 105.3, 105.7 and 106.5, respectively. The carbon-13 NMR (^13^C-NMR) and HSQC spectra of **1** confirmed the 30 aglycon signals and 23 sugar moiety signals. The ^13^C-NMR data of **1** were similar to those of 3*β*-hydroxy-30-norolean-12-en-28-oic acid (oleanolic acid) in the same NMR solvent. The absolute configuration of the sugar components was established as d-glucose, l-arabinose by high-performance liquid chromatography (HPLC) after acid hydrolysis (Section 4.4). The all-sugar proton signals were confirmed by correlation spectroscopy (^1^H-^1^H COSY) and the corresponding ^13^C resonances were then identified by HSQC (Appendix A). The sequence of the saccharide chain Glc(1→2)Ara was confirmed by the heteronuclear multiple bond correlations (HMBCs) of H-1′ (*δ*_H_ 4.94; d, *J* = 5.0 Hz) with C-3 (*δ*_C_ 89.4) and H-1′′ (*δ*_H_ 4.99; d, *J* = 7.5 Hz) with C-2′ (*δ*_C_ 81.5). The HMBC correlation between the anomeric Glc H-1′′′ (*δ*_H_ 6.22) and C-28 of the aglycon (*δ*_C_ 178.4) confirmed that *β*-d-glucopyranosyl was at C-28. Additionally, the strong HMBC signals of Glc H-1′′′′ (*δ*_H_ 5.15; d, *J* = 7.5 Hz) with C-6′′′ (*δ*_C_ 70.6) confirmed that the sugar linkages were 28-*O*-*β*-d-glucopyranosyl-(1→6)-*β*-d-glucopyranoside. Based on this evidence, the structure of compound **1** was identified as: 3-*O*-[*β*-d-glucopyranosyl (1→2)-α-l-arabinopyranosyl] oleanolic acid-28-*O*-[*β*-d-glucopyranosyl (1→6)-*β*-d-glucopyranosyl] ester [19].

Compound **2** was obtained as a white powder, with [α]d^20^ +35.0 (*c* = 1.0, MeOH). The IR spectrum of **2** showed strong absorption of hydroxyl (3398 cm^−1^) and ester (1728 cm^−1^) groups and double bonds (1066 cm^−1^). The molecular formula was identified by liquid chromatography–mass spectrometry (LC-MS) to be C_48_H_78_O_18_ from the positive-ion [M + NH_4_]^+^ at *m*/*z* 930.5. Similar to compound **1**, the ^1^H-NMR spectrum of **2** showed seven tertiary methyl groups at *δ*_H_ 0.88, 0.90, 0.91, 1.02, 1.12, 1.27 and 1.33, with 3H, singlet signals. Analysis of the chemical shifts of **2** led to the same aglycon as **1**. Three anomeric protons were observed at *δ*_H_ 4.77 (t, *J* = 4.4 Hz, H-1′), 4.99 (d, *J* = 8.0 Hz, H-1′′′) and 6.21 (d, *J* = 8.0 Hz, H-1′′), corresponding to the three anomeric carbons at *δ*_C_ 96.2, 105.4 and 107.4, respectively. The HMBC correlation of C-3 (*δ*_C_ 89.4) with H-1′ (*δ*_H_ 4.77; d, *J* = 4.4 Hz) confirmed that Ara-1 was linked at the C-3 position of the aglycon. The linkage at the C-28 position and sequence of the disaccharide chain Glc(1→6)Glc were confirmed by the HMBC correlations of H-1′′ (*δ*_H_ 6.21; d, *J* = 8.0 Hz) with C-28 (*δ*_C_ 177.2) and H-1′′′ (*δ*_H_ 4.99; d, *J* = 8.0 Hz) with C-6′′ (*δ*_C_ 69.9) (Appendix A). The absolute configuration of the sugar components was established as d-glucose, l-arabinose by HPLC after acid hydrolysis. Therefore, compound **2** was identified as 3-*O*-*α*-l-arabinopyranosyl oleanolic acid-28-*O*-[*β*-d-glucopyranosyl (1→6)-*β*-d-glucopyranosyl] ester [19]. To the best of our knowledge, this is the first time that compounds **1** and **2** have been reported for the Lardizabalaceae family.

### 2.2. Quantitative Analysis of Compounds ***1***, and ***2*** in the EtOH Extract of S. hexaphylla Leaves

Several studies have reported using liquid chromatography with tandem mass spectrometry (LC-MS/MS) to measure the content of saponins in herbal medicines for better accuracy [20]. To quantify the contents of compounds **1** and **2** in *S. hexaphylla*, an accurate method was developed using a mass spectrometer equipped with an electrospray ionization (ESI) source. First, [M + NH_4_]^+^ ions were obtained in the MS^1^ scanning mode (*m*/*z* range = 100–1500). Then, under the optimized HPLC-ESI-MS/MS conditions, compounds **1** and **2** were characterized by multiple reaction monitoring (MRM) through the transition from the parent ions at *m*/*z* 1092.5 [M + NH_4_]^+^ and 930.5 [M + NH_4_]^+^ to the main product ions at m/z 773.3 [M-2Glc + Na]^+^ and 347.3 [2Glc + Na]^+^, respectively (Figure 2 and Figure 3). The contents of **1** and **2** of 0.0072 and 0.0016 mg/g, respectively, in the ethanol (EtOH) extract of *S. hexaphylla* leaves were calculated by single-point calibrations.

### 2.3. Immunoregulatory Activities of the Isolated Compounds

#### 2.3.1. Cytotoxic Properties of Compounds **1** and **2**

Cytotoxicity is an undesirable property of immune modulators. To determine the cytotoxicity of compounds **1**–**2**, and oleanolic acid aglycon, we treated RAW 264.7 cells with serially diluted concentrations of individual molecules for 3 days and examined the numbers of cells in the culture by an MTT assay. The results showed that none of the molecules exerted noticeable cytotoxicity even at 20 μM (Figure 4).

#### 2.3.2. Inhibition of the Activation of the RAW 264.7 Cell Line by the Oleanolic Acid Saponins In Vitro

Nitric oxide (NO) is a key mediator for the inflammatory response and pathogenesis. We examined the inhibitory activities of compounds **1** and **2** for the activation of macrophage RAW 264.7, leading to the production of NO. When RAW 264.7 cells were treated for 24 h with either compound **1** or **2** before incubation with LPS, the levels of NO produced by those cells decreased in a concentration-dependent manner. The extent of inhibition caused by **2** was noticeably higher than that caused by **1** and oleanolic acid aglycon at the same concentrations (Figure 5). The difference in the extent of inhibition caused by **1** and oleanolic acid aglycon was marginal, if at all.

#### 2.3.3. Inhibition of the Activation of Macrophage by Orally Administered Oleanolic Acid Saponins In Vivo

TNF-α is a well-characterized, proinflammatory cytokine that plays a crucial role in host defense and the inflammatory response. TNF-α has been implicated in the pathogenesis of both acute and chronic inflammatory diseases. Thus, we also examined the effects of compounds **1** and **2** on the activation of macrophage in vivo. Oleanolic acid aglycon and compounds **1** and **2** were administered orally using an oral gavage technique for 4 days before macrophage activation in vivo with LPS (Figure 6). When mice were administered either compound **1** or **2** in saline (1% Tween 20) before LPS injection (i.p.) to cause sepsis, the levels of TNF-α in the blood were diminished markedly relative to those of TNF-α in the blood of sham-treated mice. In contrast, the treatment with oleanolic acid aglycon caused no significant decrease.

## 3. Discussion

Triterpene glycosides isolated from rare medicinal plants often display a wide range of pharmacological activities, such as anti-inflammatory [21], antifungal [22], antimicrobial [23], anti-cancer [24], anti-diabetic [25] and immunomodulatory [26] effects. Indeed, triterpene saponins are the key active constituents of major medicinal herbs such as *Panax ginseng* (Araliaceae), *Gynostemma pentaphyllum* (Cucurbitaceae) and *Glycyrrhiza uralensis* (Fabaceae) [27,28].

Previous studies demonstrated that the extract of *S. hexaphylla* leaves has an anti-inflammatory effect through suppression of nitric oxide synthase (iNOS) and cyclooxygenase 2 (COX-2) expression [5,15]. A phytochemical investigation of *S. hexaphylla* led to the isolation of two compounds, 3-*O*-[*β*-d-glucopyranosyl (1→2)-*α*-l-arabinopyranosyl] oleanolic acid-28-*O*-[*β*-d-glucopyranosyl (1→6)-*β*-d-glucopyranosyl] ester (**1**) and 3-*O*-*α*-l-arabinopyranosyl oleanolic acid-28-*O*-[*β*-d-glucopyranosyl (1→6)-*β*-d-glucopyranosyl] ester (**2**). Their structures were determined by the analysis of their spectroscopic data, IR, NMR, ESI-MS and chemical reaction. Furthermore, the quantitative analysis of their compounds were first time reportedly. The potential anti-inflammatory effects of isolated compounds were also reported by the in vitro and in vivo study. The structure-activity relationship (SAR) in triterpenoid saponins may be deduced from anti-inflammatory effects. Consideration of the structure-activity relationship (SAR) of these triterpenoid saponins suggested that the presence of the sugar units at C-3 and/or C-28 of the aglycon might play an important role in the anti-inflammatory inhibitory activity of these compounds. Thus, in this study, the preliminary comparison anti-inflammatory activity of these saponins with their aglycon was also discussed.

Efforts have been made to identify the specific chemical components responsible for the immunomodulatory activities of *S. hexaphylla* [5]. As a result, various oleanolic acid-based saponins, as well as its aglycon, were isolated preferentially as pure forms, suggesting that oleanolic acid-based glycosides are, at least in part, responsible for the pharmacological activities of *S. hexaphylla*.

It is well accepted that saponins have superior pharmacological activities relating to their aglycon [29]. In line with this notion, compound **2** showed a somewhat more potent inhibition of the activation of the RAW 264.7 cell line in vitro than oleanolic acid aglycon. The difference in the extent of the inhibition by **2** and oleanolic acid aglycon seemed more conspicuous at the higher concentrations (i.e., 3.3 and 10 μM) than at lower concentrations (Figure 5). Notably, however, the difference was, at most, about 15%. These results indicate that oleanolic acid aglycon can fully exert immunomodulatory activities and that the sugar constituents may have only some additive effect.

Given the results from the in vitro assay, it was rather surprising that the in vivo assay results indicated that compounds **1** and **2** exerted much stronger macrophage activation inhibition than oleanolic acid aglycon. The difference between the in vitro and in vivo assay results is not readily reconciled. However, it is postulated that the poor solubility of oleanolic acid aglycon in aqueous solution may be responsible for the poor pharmacological activities shown in the in vivo assay. It is known that the addition of sugar moieties to a triterpene compound increases its solubility. As the molecules were given orally during the in vivo assays, it is conceivable that the poor aqueous solubility of oleanolic acid aglycon resulted in poor absorption in the gastrointestinal tract.

The results that the olanolic acid saponins exerted better effects than their aglycon, when they were formulated in an aqueous solution and administered orally, have important implications.

Considering that traditional oriental medicines are generally prepared by simmering medicinal herbs in water (H_2_O) and administered orally, it is likely that saponins play more important roles than their aglycons in the actions of traditional oriental medicines. It is possible that the pharmacological properties of the sapogenin core may change depending on the composition and arrangement of the sugar moieties to be attached. Thus, further research is warranted to understand the structure-activity relationship between the sugar moieties in saponins and their pharmacological properties, particularly in vivo. Our study suggests that oleanane triterpene saponins from *S. hexaphylla* leaves could be developed as a therapeutic agent for treating inflammatory diseases.

## 4. Materials and Methods

### 4.1. Chemical and Reagents

The optical rotation values were recorded with a JASCO P-1020 polarimeter (JASCO, Tokyo, Japan). FT-IR spectra were performed on a JASCO Report 100 infrared spectrophotometer (JASCO, Tokyo, Japan). All NMR spectra were carried out on Bruker Avance III 600 spectrometers (Bruker, Billerica, MA, USA) in pyridine-*d_5_* with TMS as the internal standard. NMR data analysis was conducted with MestReNova version 12.0.4 (Metrelab Research SL, Santiago de Compostela, Spain). LC-MS/MS data were performed by using a Shimadzu LCMS-8040 system (Shimadzu, Kyoto, Japan) in positive and negative mode [14]. All solvents used for isolation were provided by the SK chemical Company, Korea. Silica gel (70-230, 230–400 mesh, Merck, Whitehouse Station, NJ, USA) and YMC RP-18 resins (75 µm, Fuji Silysia Chemical Ltd., Kasugai, Japan) were used as absorbents in the column chromatography. Thin layer chromatography (TLC) plates were purchased from Merck KGaA (silica gel 60 F_254_ and RP-18 F_254_, 0.25 µm, Merck, Darmstadt, Germany). Spots were detected under UV radiation (254 and 365 nm, Ultra-Violet Products Ltd., Cambridge, UK) and by spraying the plates with 10% H_2_SO_4_ followed by heating with a heat gun. Chemical reagents and standard compounds were purchased from Sigma-Aldrich (St. Louis, MO, USA).

### 4.2. Plant Material

The plant material *S. hexaphylla* (Lardizabalaceae) leaves were collected in the area of Jangheung, Jeollanam-do, Korea and authenticated by the National Institute of Biological Resources (Supporting data, Appendix A). A voucher specimen (CNU 17002) was deposited at the Herbarium of College of Pharmacy, Chungnam National University (Daejeon, Korea).

### 4.3. Biodirected Isolation of the Active Compounds

The dried leaves of *S. hexaphylla* (2.0 kg) were extracted with 95% aqueous EtOH (10 L × 3 times) under reflux condition. Evaporation of the solvent under reduced pressure gave the EtOH extract (700 g). The EtOH extract was suspended in H_2_O and successively separated with dichloromethane (CH_2_Cl_2_) and ethyl acetate (EtOAc) to yield the CH_2_Cl_2_ extract (300 g), EtOAc extract (210 g) and H_2_O layer, respectively. Low-range polarity fraction (DCM and EtOAc extracts, notably contained chlorophyll, essential oil and fatty substances). Furthermore, the H_2_O fraction showed potent NO inhibitory activity, which was greater than that in the presence of the EtOH extract. Thus, this H_2_O fraction of *S. hexaphylla* leaves was chosen for subsequent study.

The H_2_O layer was separated using a Diaion HP-20 column and was eluted with a gradient solvent mixture of MeOH-H_2_O (from 25:75 to 50:50, 75:25 and pure MeOH, stepwise) to yield four fractions (W-1 to W-4, respectively), based on the thin-layer chromatography (TLC) analysis. The fraction W-4 showed stronger NO inhibitory activity than the fractions W-1-W-3. Thus, fraction W-4 was chosen as the focus for active compound isolation. Fraction W-4 was subjected to silica gel column chromatography (CC), eluting with the gradient solvent system of CH_2_Cl_2_/MeOH/H_2_O (0–100% of MeOH, stepwise) to obtain eight fractions (W-4.1 through W-4.8). Fraction W-4.2 was separated by YMC RP-18 CC with acetone/H_2_O (1.5/1. *v*/*v*) as an eluent to give four small subfractions. Fraction W-4.8 was isolated by silica gel CC eluting with CH_2_Cl_2_/MeOH/H_2_O (2/1/0.2. *v*/*v*). Purification with Sephadex LH-20 provided compounds **1** (400.6 mg) and **2** (200.8 mg).

Physical spectroscopic data of the isolated compounds: 3-*O*-[*β*-d-glucopyranosyl (1→2)-*α*-l-arabinopyranosyl] oleanolic acid-28-*O*-[*β*-d-glucopyranosyl (1→6)-*β*-d-glucopyranosyl] ester (**1**). White amorphous powder; [α]d^20^+ 7.05 (c 1.5, MeOH). IR (KBr) *ν_max_* 3398, 2988, 1728, 1066 and 1046 cm^−1^. ^1^H (600 MHz, pyridine-*d_5_*) and ^13^C-NMR (150 MHz, pyridine-*d_5_*) data, see supporting data, (Appendix A); ESI–MS *m*/*z* 1092.5 [M + NH_4_]^+^ and 1119.5 [M + HCOO^−^]^−^.

3-*O*-*α*-l-arabinopyranosyl oleanolic acid-28-*O*-[*β*-d-glucopyranosyl (1→6)-*β*-d-glucopyranosyl] ester (**2**). White amorphous powder; [α]d^20^ + 35.0 (c 1.0, MeOH). IR (KBr) *ν_max_* 3398, 2988, 1728, 1066 and 1046 cm^−1^. ^1^H (600 MHz, pyridine-*d_5_*) and ^13^C-NMR (150 MHz, pyridine-*d_5_*) data, see Appendix A; ESI-MS *m*/*z* 930.5 [M + NH_4_]^+^ and 957.4 [M + HCOO^−^]^−^.

### 4.4. Acid Hydrolysis and Sugar Identification

Compounds **1** and **2** (2.0 mg) were dissolved in 1 mL of 1.0 N HCl (H_2_O/dioxane, 1:1, *v*/*v*), and each solution was heated at 80 °C for 8 h. After cooling, the solvent was evaporated under N_2_, and the residue was separated by solvent–solvent partition using CH_2_Cl_2_ and water. The solvent system was CH_2_Cl_2_/MeOH/H_2_O 2/1/0.2, and spots were visualized by spraying with 95% EtOH-H_2_SO_4_-anisaldehyde (9/0.5/0.5, *v*/*v*), then heated at 200 °C for 7 min. The *R*_f_ values of glucose, and arabinose detected by TLC were 0.30 and 0.50, respectively. The water layer was concentrated under N_2_, and the remaining portion was dissolved in 3 mL of pyridine and heated with 10 mg of l-cysteine methyl ester hydrochloride. After heating at 80 °C for 60 min, 0.1 mL of *o*-tolylisothiocyanate was reacted with the mixture at 80 °C for 60 min. The reaction mixture evaporated under vacuum. The reaction mixtures were directly analyzed by reversed-phase HPLC (Phenomenex column 250 mm × 4.6 mm, 5 µm), with a mobile phase MeCN-H_2_O (25–75); flow rate of 0.5 min/mL; UV detection at 250 nm. The derivatives of d-glucose, and l-arabinose in compounds **1** and **2** were determined by comparison to the retention times of the standards (*t*_R_: l-arabinose 15.6 min, and d-glucose 28.8 min).

### 4.5. LC-MS/MS Analysis of the Isolated Compounds

LC-MS/MS analysis were performed by using a Shimadzu LCMS-8040 system (Shimadzu, Kyoto, Japan) in positive and negative mode. Electrospray (Shimadzu, Kyoto, Japan) was operated under the following condition, the interface voltage was −3.5 kv for the negative mode and the positive mode was 4.5 kv, the nebulizing gas flow rate was 3 L/min, drying gas flow rate was 15 L/min, a desolvation line temperature was 250 °C, and the heat block temperature was 400 °C. The stationary phase was Hector M 18 (250 mm × 4.6 mm, 5 μm, RSTech, Daejeon, Korea). The mobile phase consisted of (A) 0.1% formic acid and (B) acetonitrile-formic acid (99.9:0.1) using a gradient elution of 28–50% B at 0–60 min under the flow rate of 0.5 mL/min. The optimized collision energy in the MRM mode for compounds **1**–**2** were 18 V and 22 V, respectively.

### 4.6. Biological Studies

#### 4.6.1. Cytotoxicity Assay Using MTT

RAW 264.7 cells were maintained in DMEM medium with 10% fetal bovine serum (FBS) in a humidified CO_2_ incubator. Cell cytotoxicity was evaluated using the 3-(4, 5-dimethylthiazol-2-yl)-2, 5-diphenyl tetrazolium bromide (MTT) assay. Briefly, RAW264.7 cells suspended in DMEM medium (10% FBS, Sigma-Aldrich, St. Louis, MO, USA) were seeded onto a 96-well culture plate at a density of 5 × 10^3^ cells/100 μL. The next day, 100 µL of the fresh medium was supplemented to the culture, and then 2 µL of compounds **1** and **2**, oleanolic acid and dexamethasone in DMSO were added to the culture at 20 μM, 5 μM, 1.25 μM, 0.32 μM and 0.08 μM, respectively. Three days (72 h) after culture with the compound at 37 °C, MTT assays were performed as described previously. The assays were run in triplicate.

#### 4.6.2. Nitric Oxide Production Assay with the RAW 264.7 Macrophage Cell Line

RAW 264.7 cells were plated at 1 × 10^4^ cells/well in a 96-well plate and were incubated overnight. Then, compound **1**, compound **2**, oleanolic acid and dexamethasone were added to the cultures at 10 μM, 3.3 μM, 1.1 μM, 0.37 μM and 0.12 μM, respectively. After 1 and 24 h culture with each compound at 37 °C, respectively, LPS was supplemented to the cultures at 0.5 μg/mL. After 24 h culture at 37 °C, nitric oxide concentrations in the culture supernatants were measured using NO detection kit (21023, LiliF Diagnostic, Seoul, Korea)

#### 4.6.3. LPS-Induced Sepsis and Measurement of TNF-α in the Blood

The experimental procedures were approved (Approval number: 202003A-CNU-062) by the Animal Ethics Committee in Chungnam National University, and animal experiments were carried out per the approved protocols.

For animal experiments, all of the compounds were first pulverized with Tween 20 and then diluted with saline (1:99) to maximize the solubility of compounds in water [30]. The compounds in the saline/Tween 20 solution (2.5, 0.83 and 0.28 mg/mL, respectively) was administered via oral gavage (200 µL per injection). Mice (male BALB/C, 8 weeks; Samtako, Osan, Korea) were pretreated orally with individual compounds for 4 days. Two hours after the last treatment, the mice were injected i.p. with LPS (0.1 mg/kg) [31]. Ninety min after the LPS injection, the blood was collected, and the concentration of TNF-α in the blood was measured with TNF-α ELISA kit (Abcam, Cambridge, MA, USA). The experiment was performed with 3 mice per group.

#### 4.6.4. Statistical Analysis

The significance of the difference among separate experimental groups was determined with a nonparametric one-way ANOVA (Kruskal-Wallis test), and Bonferroni correction was used as post hoc analysis. *p* < 0.05 was considered statistically significant.

## Figures and Tables

**Figure 1 molecules-25-03699-f001:**
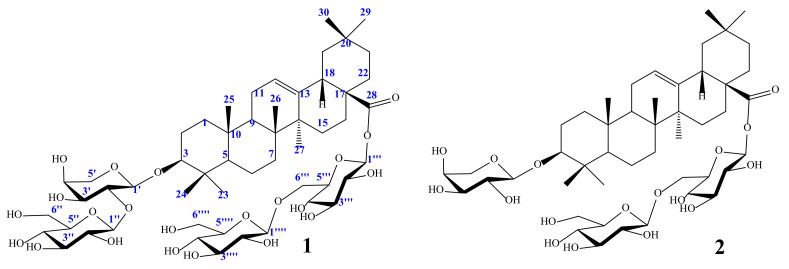
The structure of the isolated compounds **1** and **2**.

**Figure 2 molecules-25-03699-f002:**
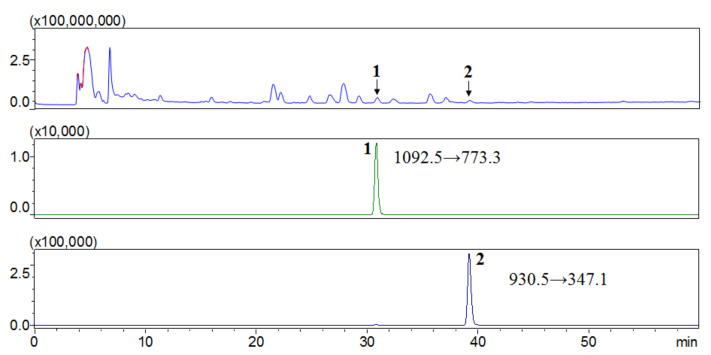
Positive ion mode LC-electrospray ionization (ESI)-MS/MS multiple reaction monitoring (MRM) analysis of EtOH extract of *Stauntonia hexaphylla* leaves. Profile of the specific transition of compounds **1** and **2** from precursor ion to product ions in the MRM mode.

**Figure 3 molecules-25-03699-f003:**
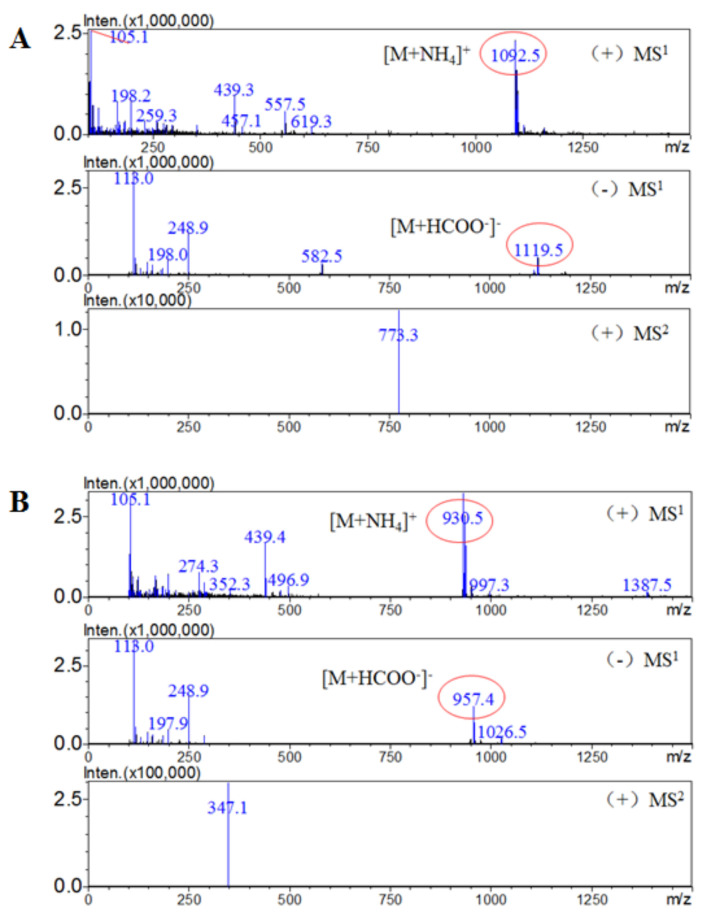
MS^1^ and MS^2^ spectrum of compounds **1** (**A**) and **2** (**B**).

**Figure 4 molecules-25-03699-f004:**
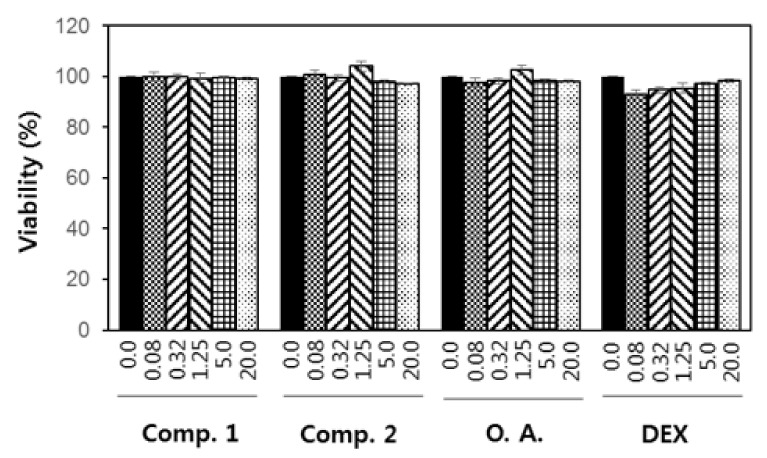
Cytotoxic properties of compounds **1** (Comp. **1**) and **2** (Comp. **2**) RAW 264.7 cells after treatment with compounds **1**–**2**, oleanolic acid (**O.A**.) and dexamethasone (**DEX**) (0.08, 0.32, 1.25, 5 and 20 µM, respectively) for 3 days. Dexamethasone was used as the positive control.

**Figure 5 molecules-25-03699-f005:**
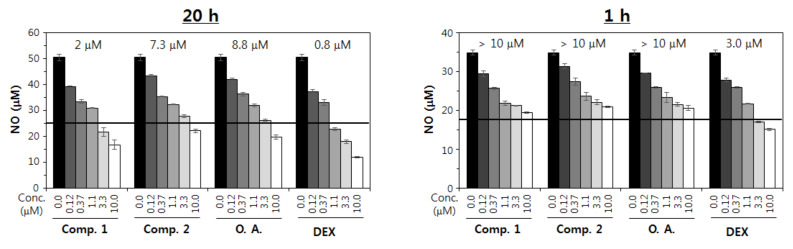
Inhibition of the activation of the RAW 264.7 cell line by the active compounds **1** (Comp. **1**) and **2** (Comp. **2**), in vitro. Cells were treated with different concentrations of compounds **1**–**2**, oleanolic acid (**O.A.**) and dexamethasone (**DEX**), separately. Nitrite oxide concentrations in the culture supernatants were determined using a Griess reagent assay. Dexamethasone was used as the positive control.

**Figure 6 molecules-25-03699-f006:**
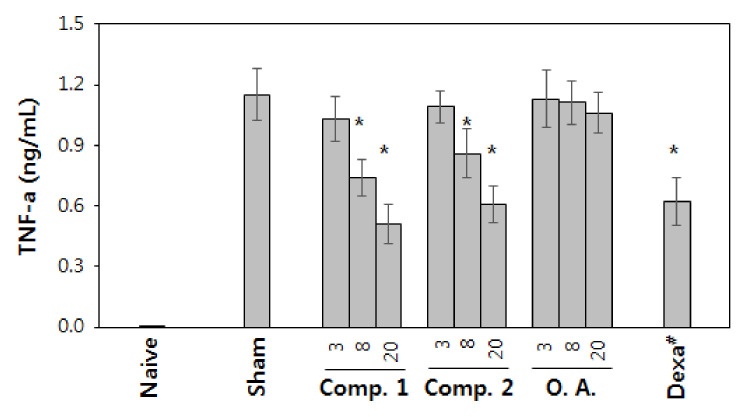
Inhibition of the activation of macrophage by orally administered compounds in vivo. After 4 days treatment with 20 mg/kg compounds **1** (Comp. **1**) and **2** (Comp. **2**) and oleanolic acid (**O.A.**), respectively, LPS was injected i.p., and the blood was drawn for the analysis of the concentration of TNF-α. The concentration of TNF-α was determined with ELISA (* *p* <0.05, ANOVA). Dexamethasone (**Dexa^#^**) was used as the positive control. Sham means a negative control group that is injected with LPS to induce inflammation but not treated.

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
