# Peer review of "Enhancement of an In Vivo Anti-Inflammatory Activity of Oleanolic Acid through Glycosylation Occurring Naturally in Stauntonia hexaphylla"

_molecules, 2020, doi:10.3390/molecules25163699_

Round 1
Reviewer 1 Report
The authors have taken into consideration most of the comments of the referee however, there remain minor points which needed to be taken into account
L 56 oleane------>oleanane. Could you check again all the manuscript in this regard?
L64 comparison------>comparison of
L112, 187, 259, 282
saponin compounds------>, check all the manuscript in this regard
122 GLc------->Glc
Author Response
Covering Letter to Reviewers’ Comments on Original Manuscript
Ms. No.: Molecules-850419
Title: Enhancement of an in vivo anti-inflammatory activity of oleanolic acid through glycosylation occurring naturally in Stauntonia hexaphylla
Corresponding Author: Professor Seo Young Yang and Professor Inkyu Hwang
Authors: Le Ba Vinh; Nguyen Thi Minh Nguyet; Liu Ye; Gao Dan; Nguyen Viet Phong; Hoang Le Tuan Anh; Young Ho Kim and Jong Seong Kang.
Reviewer #1: Comments and Suggestion for Authors
The authors have taken into consideration most of the comments of the referee however, there remain minor points which needed to be taken into account
Response: We would like to thank the Reviewer for taking the time and effort necessary to review the manuscript. We sincerely appreciate all the valuable comments and suggestions, which helped us to improve the quality of the manuscript.
L 56 oleane------>oleanane. Could you check again all the manuscript in this regard?
Response: Thank you for your valuable comment. The authors have carefully checked the errors. We changed “oleane” to “oleanane” in the revised manuscript.
L64 comparison------>comparison of
Response: We corrected the error as ‘comparison of’ in the revised manuscript.
L112, 187, 259, 282
saponin compounds------>, check all the manuscript in this regard
Response: The authors have corrected the names in the revised manuscript.
122 GLc------->Glc
Response: We changed ‘Glc’ in the revised manuscript.
Reviewer 2 Report
The revised manuscript is fine.
I recommend this manuscript is to be published in Molecules.
Author Response
Covering Letter to Reviewers’ Comments on Original Manuscript
Ms. No.: Molecules-850419
Title: Enhancement of an in vivo anti-inflammatory activity of oleanolic acid through glycosylation occurring naturally in Stauntonia hexaphylla
Corresponding Author: Professor Seo Young Yang and Professor Inkyu Hwang
Authors: Le Ba Vinh; Nguyen Thi Minh Nguyet; Liu Ye; Gao Dan; Nguyen Viet Phong; Hoang Le Tuan Anh; Young Ho Kim and Jong Seong Kang.
Reviewer #2: Comments and Suggestion for Authors
The revised manuscript is fine.
I recommend this manuscript is to be published in Molecules.
Response: The authors would like to thank you for accepting our manuscript for publication in the Journal.
This manuscript is a resubmission of an earlier submission. The following is a list of the peer review reports and author responses from that submission.
Round 1
Reviewer 1 Report
This manuscript described the isolation, structure elucidation and anti-inflammatory activity of two saponins isolated from Stauntonia hexaphylla.
There are too many inconsistencies in this manuscript
Firstly the phytochemical part has already been published in Bioorganic medicinal chemistry letters 2019, 29, 965 the described compounds 1, 2 were already described a compounds 10 and 23 in the previous paper .
Secondly the in vitro test with assessment of the effect of 1, 2 on the NO secretion in RAW 264.7 cells activated by LPS has already been published in the same article showing IC50 >50 microM
Here you annonce a dose-dependent effect , and you have to express the results as IC50
Thirdly, The in vivo assay described in this manuscript is the only part which has not been previously published , but this assay doesn’t express the significance of the results by comparison with a positive control and this experiment which has been achieved with only one dose is not sufficient.
Therefore the title is absolutely not realistic
Throughout the manuscript the authors have written “oleane saponins ” instead of “oleanane saponins”
some Latin names are not italicized
Reviewer 2 Report
The manuscript deals with isolation and identification of two known oleane saponins from Stauntonia hexaphylla (fam. Lardizabalaceae). Both compounds have been tested for in vitro anti-inflammatory activities.
Please, find my comments in the attached file.

Reviewer 3 Report
Authors reported the structures and anti-inflammatory activity of oleane triterpene saponins. Those compounds apparently inhibited the activation of macrophage. I recommend this manuscript is to be published in Molecules.
1) To confirm the molecular weight, high resolution MS should be shown.
2) Page 2, Line 70. Correct proton numbers of aromatic protons: 6.22 (H-1’), 4.94 (H-1’), 4.99 (H-1’).
3) Page 2, Line 82, H-1”” (dH 5.15; d, J=7.5 Hz), but on line 71, 5.15 (d, J=7.5 Hz, H-1”) Confirm the proton assignment.
4) The ester linkage H-1’’’/C-28 in compound 1 was not described. Was HMBC H-1’’’/C-28 observed?
5) Page 3, Figure 1. The structure of arabinose is drawn as a chair form. The direction of oxygen (1’-O-3) looks axial and on the same plane of 1’-O-5’. Please draw structures clearly.
6) Page 4, Figure 3. In MS/MS spectra, different product ions, 773.3 and 347.1, were observed from compound 1 and 2, respectively. The position of cleavage sites is not clear.
Compounds 1 and 2 are analogs. Why were the different product ions observed?
7) In Figure 2, the product ion of 2 is 347.3 but in Figure 3 the product ion is observed at 347.1.
8) In supplemental material. The scale of proton chemical shifts in HMBC (Figure S6) is wrong.
Reviewer 4 Report
Manuscript Molecules 850419 describes the study 2 compounds isolated from the plant Stauntonia hexaphylla. After isolation and characterisation by NMR and HPLC analyses, anti-inflammatory in vitro experiments using cell cultures (RAW 264.7) and in vivo mice were carried out. Cell culture experiments include viability with MTT assay and determination of nitric oxide. Animal experiments consisted in oral administration of the compounds for 4 days and TNF-alpha analysis in blood.
The article is incomplete, not clear written and not well organise. Morevoer, English must be improved. Therefore, I don’t recommend its publication in its present form.
Some recommendations for authors to improve the article are as follows:
ABSTRACT:
Improve abstract by describing the main results; indicate which animal model has been used…Maybe indicate here saponins 1 and 2 to save space for the rest of information for abstract. Include a last phrase with conclusions.
INTRODUCTION
It is very short and does not explain the relevance of the study. Have other saponins described as anti-inflammatory? Is there any mechanism of anti-inflammation relevant for the study?
Lines 41-48 do not add any information. I would indicate that NO and TNF-alpha are mediators of inflammation and used to test potential anti-inflammatory properties of compounds.
The aim of the article is not clear. Please include a statement with the aim of the article.
RESULTS
They are better written.
Include statistic in Figure 5 and indicate number of replicates/plates carried out.
DISCUSSION
It is very short.
There is a lack of references from other authors in relation to the terpenes and the potential anti-inflammatory activity of these compounds.
Lack of conclusions.
MATERIALS AND METHODS
This section is poorly described.
Subsection 4.1 should describe a list of the most important reagents and chemicals used in the study and not the description of the equipment. The latter where the equipment is used, i.e, in the HPLC analysis of compounds.
Subsection 4.3 lines 213-214. The NO inhibitory activity has not described in this point. Is it the Nitric oxide production assay with cell cultures?
Subsection 4.4 Indicate volume of injection and concentration of the sample.
Subsection 4.5.1. Where RAW264.7 cells were purchased/donated? Line 247 is DEME medium a typo and should be DMEM or DME or MEM? Why cells were incubated with oleanolic acid and dexamethasone? Where they controls? What is the final concentration of DMSO in the cell studies? How long were cells incubated with the compounds, 72h? How many plates and replicates?
Section 4.5.2 improve the description of the experiment was the cells washed before adding LPS? Times of incubation?
Section 4.5.3 How many mice were used? How many were included in the control group? Why Tween 20 was used to dissolved the compounds? What is the concentration amount of compound administered?